# Disease-related and age-related changes of anterior chamber angle structures in patients with primary congenital glaucoma: An in vivo high-frequency ultrasound biomicroscopy-based study

**Yan Shi[1], Ying Han[2], Chen Xin[1], Man Hu[3], Julius Oatts[2], Kai Cao[1], Huaizhou Wang[1]☯\*, Ningli Wang** **[1]☯\***

**1** Beijing Tongren Eye Center, Beijing Tongren Hospital, Beijing Institute of Ophthalmology, Capital Medical University, Beijing, China, **2** Department of Ophthalmology, University of California, San Francisco School of Medicine, San Francisco, CA, United States of America, **3** Department of Ophthalmology, National Key Discipline of Pediatrics, Ministry of Education, Beijing Children's Hospital, Capital Medical University, Beijing, China

☯ These authors contributed equally to this work.
\* wningli@vip.163.com (HW); trhz_wang@163.com (NW)

**Data Availability Statement:** All relevant data are within the manuscript and its Supporting Information files.

## Abstract

### Objectives

To provide in vivo measurements of anterior chamber angle structures and their relationship with age as evaluated by high-frequency ultrasound biomicroscopy (UBM) in patients with primary congenital glaucoma (PCG)

### Methods

High-frequency UBM was done for 51 PCG eyes from 40 patients (aged from 3 to 96 months) and 11 unaffected contralateral eyes. Parameters, including the proportion of observable abnormal tissue membrane and Schlemm's canal, the largest cross-sectional area (CSA) of Schlemm's canal (SC), SC meridional diameter, trabecular-iris angle (TIA), trabecular meshwork (TM) thickness, iris thickness, ciliary process length, and corneal limbus thickness were compared between the two groups and their relationship with age was explored in PCG eyes.

### Results

Abnormal tissue membrane was detected in 27.5% of PCG eyes and none in unaffected eyes. SC was observed in 73.1% of PGC eyes compared to 100% in unaffected eyes (P<0.001). The largest CSA of SC, SC meridional diameter, iris thickness, and corneal limbus thickness were all significantly smaller in PCG eyes compared to unaffected eyes (all P<0.05). TIA and ciliary process length in unaffected eyes were smaller than PCG eyes

**Funding:** Yan Shi received fundings from Beijing Municipal Science & Technology Commission (No. Z181100001718044) and the priming scientific research foundation for the junior researcher in Beijing Tongren Hospital, Capital Medical University (No.2018-YJJ-ZZL-028). The funders had no role in study design, data collection and analysis, decision to publish, or preparation of the manuscript.

**Competing interests:** The authors do not have competing interests.

(both $P<0.05$). The largest CSA of SC, TM thickness, iris thickness, and ciliary process length were all significantly correlated to age in PCG eyes ($P<0.05$).

## Conclusions

The anatomical information evaluated by high-frequency UBM may provide glaucoma specialists a useful tool to aid in understanding the dysgenesis and changes with age of anterior chamber angle in PCG.

## Introduction

Primary congenital glaucoma (PCG) is the most common types of the childhood glaucoma and is thought to be secondary to developmental defects in Schlemm's canal (SC) and the trabecular meshwork (TM), the main aqueous humor outflow structures [1]. *In vivo* physiological features of these structures would provide insight into the ocular structures involved in PCG; however, previous studies of SC and TM in PCG have been limited to histologic specimens *in vitro* [2–4].

The invention of ultrasound biomicroscopy (UBM) provides a noninvasive, real-time, dynamic, continuous, and in vivo assessment of the morphology of the anterior segment structure in PCG patients, and more importantly, it is suitable for examination in infants and children under general anesthesia in the supine position [5–8] compared with standard anterior segment optical coherence tomography (OCT), which can only be performed in a seated, awake child. Standard 50 MHz UBM is limited by its low resolution and cannot provide quantitative measurements of SC and TM [5,6,9]. A newer 80 MHz high-frequency UBM, the iUltrasound imaging system (iScience Interventional Inc., Menlo Park, CA), affords a more detailed view of the anterior segment than previous 50 MHz UBM with an improvement in axial resolution. Quantitative assessment of SC and TM in adults were recently reported, [10–12] and its application in PCG was limited to one published report about the measurement of SC diameter in a small case series [13]. Here, we measured *in vivo* anterior chamber angle structure dimensions and evaluate their relations to age in patients with PCG using high-frequency UBM, which are essential in understanding the pathogenesis of congenital glaucoma [6–8,13–15].

## Materials and methods

This prospective study adhered to the tenets of the Declaration of Helsinki and was approved by the Ethics Committee of Beijing Tongren Eye Center. Each patient's legal guardian or representative signed an informed consent. The clinical trial was registered under the Chinese Clinical Trials Registry (ChiCTR-OCC-15005789).

### Subjects

All patients with newly diagnosed PCG between February 2015 and March 2018 at Beijing Tongren Eye Center were enrolled. Patients underwent complete ophthalmologic examination under general anesthesia. The diagnosis of PCG was based on the presence of at least 2 of the following clinical features: (1) increased corneal diameter ($>12$ mm) together with elevated intraocular pressure (IOP) ($>21$ mm Hg), (2) Haab's striae, (3) corneal edema and (4) increased cup-to-disc ratio. Exclusion criteria included other ocular or systemic anomalies or

any previous intraocular surgery. Those with severe trabeculodysgenesis under high-frequency UBM, corresponding to those with insertion of both the iris and ciliary processes before the scleral spur as we previously reported, were excluded due to the difficulty in identifying the scleral spur and measuring TM [15]. And those with no identified SC in all four quadrants under high-frequency UBM were also excluded.

IOP was measured using the Icare tonometer (Icare TA01i, Icare Finland Oy, Espoo, Finland). For patients with bilateral disease, both eyes were included in the analysis (Group 1). For those with unilateral disease, the unaffected contralateral eyes served as the control group (Group 2). Parameters including gender, age, IOP, and corneal diameter were recorded.

## Imaging of the anterior chamber angle

The anterior chamber angle was examined in the supine position under general anesthesia using the iUltrasound imaging system by the same investigator (YS). A low-viscosity gel was placed to assist in transduction. The self-contained probe was placed directly on the eye for imaging in the 3, 6, 9, and 12 o'clock meridians. Images were obtained using the following settings: transducer frequency, 80 MHz; axial resolution, 25 μm; lateral resolution, 50 μm; electronic resolution, 10 μm; tissue penetration depth, 2 mm; scan rate, 7 frames/second; and imaging window size, $4.5 \times 4.5$ mm. In each subject, at least 20 ultrasound images for each position were obtained.

## Image processing

Two investigators (YS, CX) independently identified Schlemm's canal (SC), and the scleral spur (SS). When the investigators disagreed on the delineation of SC, a mutual conclusion was reached after discussion. These parameters were then measured by a masked, experienced investigator (YS) using ImageJ software (version 1.47, National Institutes of Health, Bethesda, Maryland, USA) transformed from pixel to anatomic values as previously described [16]. SS was defined as the end point of the curved interface between the ciliary body and the sclera/TM based on previous studies [17]. SC was defined as observable when a thin, black, lucent space adjacent to the SS and considered detected when observed on two consecutive images [12]. Presence or absence of an abnormal tissue membrane in each quadrant was recorded for analysis (Fig 1).

Details of the measurements evaluated are depicted in Fig 2. SC area was calculated using the automated area function in ImageJ. Measurements of the cross-sectional area (CSA) of SC were taken at four different positions (the 3, 6, 9, and 12 o'clock meridians), the largest of which was used for analysis to account for any variability related to visualization of SC. The percentage of eyes with observable SC was calculated for each quadrant (superior, inferior, nasal, temporal). SC meridional diameter was defined as the distance between the most posterior and anterior part of SC directly adjacent to the TM [12].

Considering the retrodisplacement of SC in PCG, the thickness of TM was measured at the anterior end point of SC, as more posterior measurements of TM might not truly represent the thickness of the TM itself but rather the ciliary muscle behind the scleral spur [3,12,18]. Iris thickness was measured across a vertical line which was perpendicular to the posterior iris plane located 500 μm centrally from the iris root. The trabecular-iris angle (TIA) was measured as the angle between the arms passing through a point on the inner surface of trabecular meshwork 500 μm from the scleral spur and the point perpendicularly opposite on the iris [9]. Ciliary process length (CLP) was measured along the line starting from the point of most-anterior tip of the ciliary body to the beginning of the zonules [6]. Corneal limbus thickness (CLT) was measured from scleral spur to the outer surface of the corneal limbus, perpendicular to the

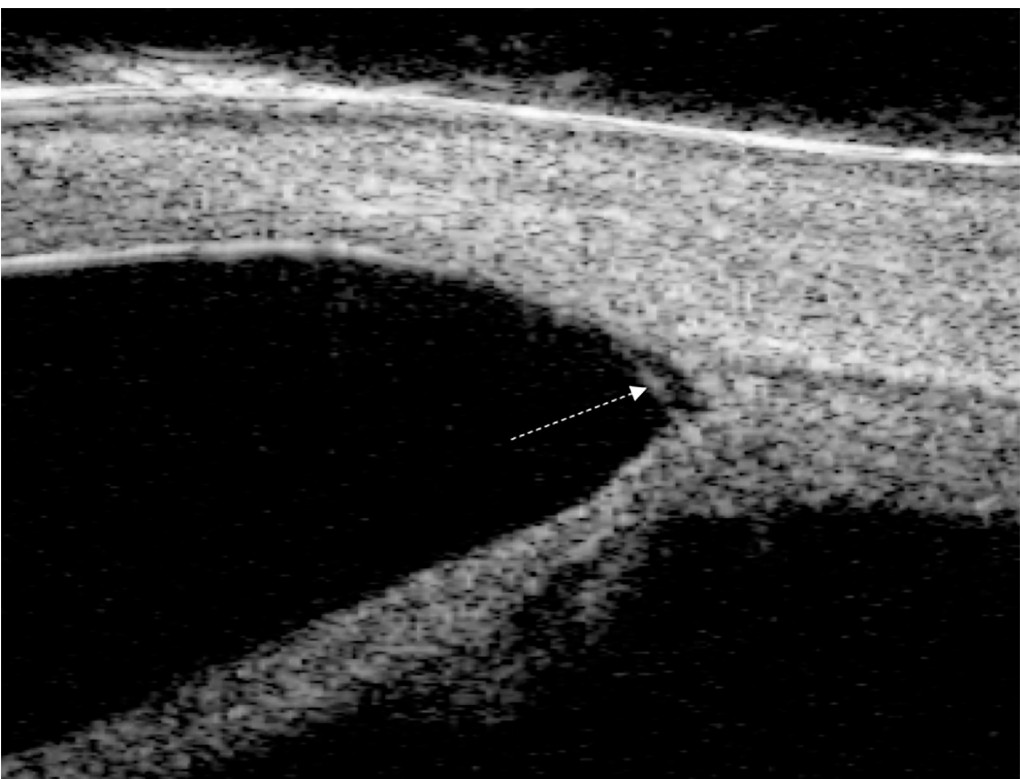

**Fig 1. Angle of a primary congenital glaucoma eye with presence of abnormal tissue membrane (dotted arrow) and absent Schlemm's canal.**

limbus tangent. Apart from the CSA of SC, measurements of other parameters were all taken at four quadrants on the image with largest SC and the average values of other anterior chamber angle parameters were used for analysis.

## Statistical analysis

All statistical analyses were performed using SPSS (V.16.0; SPSS, Chicago, Illinois, USA) or GraphPad Prism (V.7; GraphPad Software, Inc. La Jolla, CA, USA) with $p < 0.05$ considered significant. Frequency histograms and the one-sample Kolmogorov–Smirnov test were used to assess the distribution of numerical data for parametric characteristics. Qualitative and categorical data were counted by frequency, while the median (range) was used for those quantitative data which did not obey normal distribution. The Pearson $\chi 2$ test was used to compare right and left eyes between groups. A linear mixed model was used to adjust for the correlation between eyes in patients with bilateral disease and compare various parameters between Groups 1 and 2 including age, gender, IOP, corneal diameter, the distribution of quadrants with the largest SC, the proportion of observable Schlemm's canal, and other anterior chamber angle parameters. Generalized Estimating Equations was used to analyze the precise relation of age to anterior chamber angle parameters and the relation of IOP to the largest area of SC. Fifty percent of eyes (31 eyes) were randomly selected to assess observer variability, intraobserver and interobserver variability using a coefficient of repeatability and 95% limits of agreement and intraclass correlation coefficients (ICC). Interobserver agreement was calculated by comparing initial values of Observer 1 (YS) to those of Observer 2 (CX).

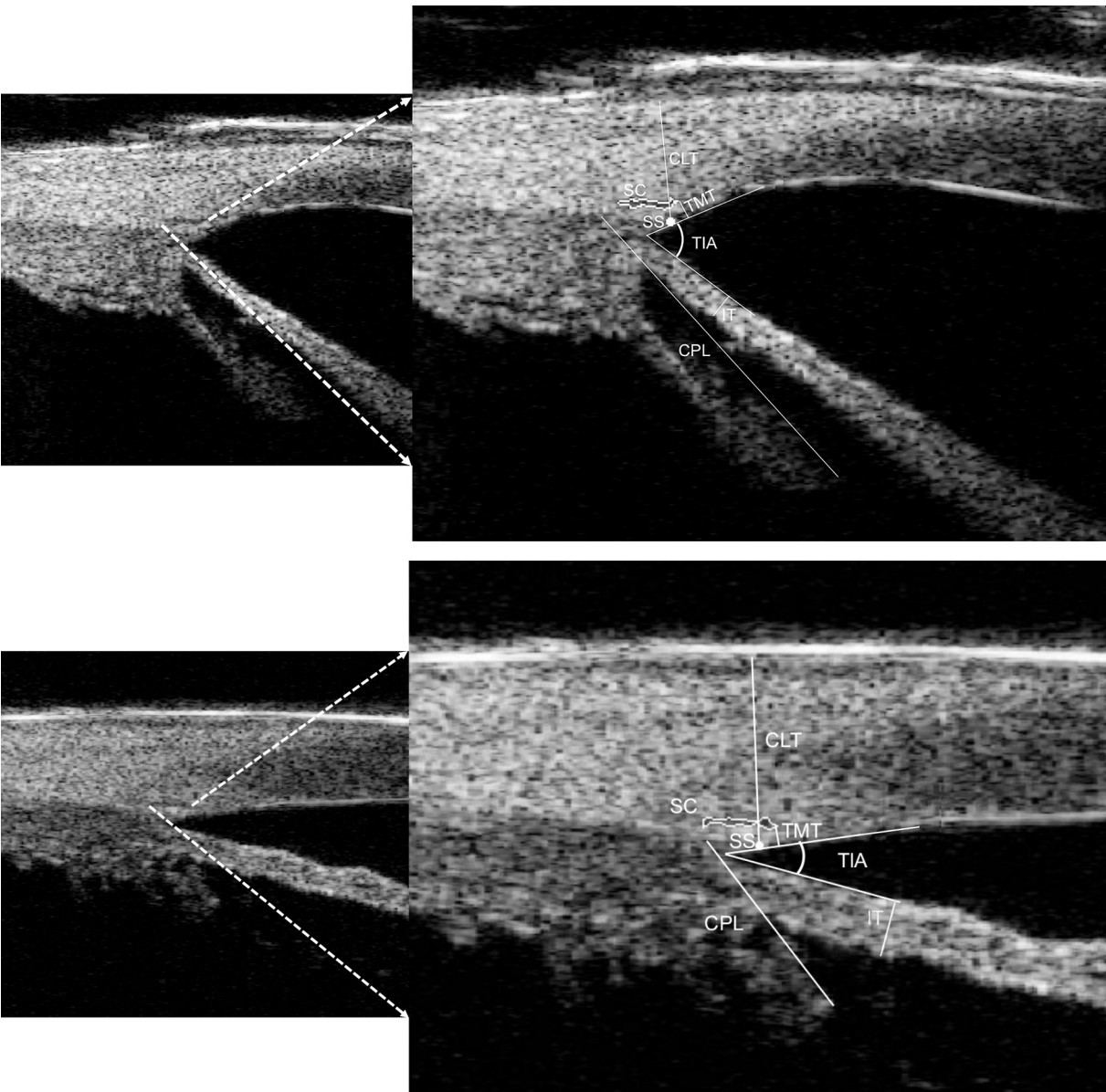

**Fig 2. High-frequency ultrasound biomiscroscopy image showing Schlemm's canal (SC) and scleral spur (SS).** Representative schematic lines and angles are shown including standard measurements of the ciliary process length (CPL), iris thickness (IT), trabecular meshwork thickness (TMT), corneal limbus thickness (CLT), and trabecular-iris angle (TIA, white angle) as previously described in other studies. Upper: image of an affected eye of a 40 months old child with primary congenital glaucoma (PCG); Lower: image of an unaffected contralateral eye of a 26 months old child with PCG.

## Results

### Subject characteristics

Fifty-one patients with 75 eyes underwent the UBM measurement, while 13 eyes were excluded for either having severe trabeculodysgenesis (4 eyes) or having no identified SC (4 eyes), or both (5 eyes). Therefore, a total of 40 patients with 62 eyes were included: 51 eyes with PCG and 11 unaffected contralateral eyes. Thirty-one patients (78%) were male. The median age was 36 months (range: 3–96). The ratio of unilateral/bilateral disease was

approximate 1:2.6 with 11 unilateral and 29 patients with bilateral PCG. There were 22 patients (38 PCG eyes and 6 unaffected contralateral eyes) had both eyes and 18 patients (13 PCG eyes and 5 unaffected contralateral eyes) had one eye recruited in this study. Comparison of subject characteristics between Group 1 and Group 2 are presented in Table 1. The number of eyes in Group 2 was significantly lower due to the lower incidence of unilateral PCG, while age, gender, and laterality was matched in the two groups (all P>0.05). The IOP and corneal diameter were significantly greater in Group 1 (both p<0.001).

## Qualitative anterior chamber angle parameters

The largest CSA of SC were identified more frequently in the nasal and inferior quadrants in Group 1 (P = 0.036) but equally distributed in Group 2 (P = 0.178) with no significant difference between groups (P = 0.686). SC was observed in 73.1% of eyes with PCG compared to 100% in unaffected contralateral eyes (P<0.001). The major difference was noted in the superior quadrant (57.7% in PCG versus 100% in normal eyes, P = 0.008) while all other quadrants had similar visibility of SC between groups (all P>0.05) (Table 1). In affected eyes, the number of eyes with abnormal tissue membrane in superior, nasal, inferior, and temporal quadrants were 11, 7, 6, 9, respectively (P = 0.545). The total percentage of the presence of abnormal tissue membrane in all quadrants was 16.2%, and the percentage of PCG eyes with abnormal tissue membrane in any quadrant was 27.5% (14 in 51 eyes). No abnormal tissue membrane was detected in unaffected eyes.

## Intraobserver and interobserver reproducibility of quantitative anterior chamber angle measurements

Analysis of the reproducibility of quantitative measurements from a randomly selected subset of 31 eyes using the intraclass correlation coefficient (ICC) is shown in Table 2. All quantitative measurements of the anterior chamber angle had good reproducibility.

**Table 1. Comparison of subject characteristics and observable Schlemm's canal proportion between groups.**

| Group | Group 1 | Group 2 | P |
|---|---|---|---|
| No. of eyes (N, %) | 51 | 11 | <0.001* |
| OD/OS | 27/24 | 6/5 | 0.595 |
| Gender (male/female) | 39/12 | 8/3 | 0.853 |
| Age(months; median, range) | 36 (3–96) | 26 (5–79) | 0.858 |
| IOP (mmHg; median, range) | 33 (22–48) | 15 (12–20) | <0.001* |
| Corneal diameter (mm; median, range) | 13.0 (12.0–16.0) | 11.1 (10.5–11.5) | <0.001* |
| Proportion of eyes with the largest CSA of SC in each quadrants (superior/nasal/inferior/temporal, %) | 9.8/35.3/33.3/21.6 | 0/54.5/36.4/9.1 | 0.686 |
| Observable SC proportion | - | - | - |
| Total (n,%) | 152/204 (74.5%) | 44/44 (100%) | <0.001* |
| Superior region (n,%) | 30/51 (58.8%) | 11/11 (100%) | 0.008* |
| Nasal region (n,%) | 40/51 (78.4%) | 11/11 (100%) | 0.142 |
| Inferior region (n,%) | 40/51 (78.4%) | 11/11 (100%) | 0.095 |
| Temporal region (n,%) | 42/51 (82.4%) | 11/11 (100%) | 0.137 |

* Statistical significance; Group 1, primary congenital glaucoma eyes; Group 2, unaffected contralateral eyes. OD, right eye; OS, left eye; IOP, intraocular pressure; CSA, cross-sectional area; SC, Schlemm's canal.

**Table 2. Reproducibility of quantitative measurements of the anterior chamber angle in a randomly selected subset of 31 eyes.**

| Anterior chamber angle parameters | Intraobserver repeatability | | | Interobserver reproducibility | | |
|---|---|---|---|---|---|---|
| | Mean | Difference | ICC (lower 95% CI) | Mean | Difference | ICC (lower 95% CI) |
| Largest area of SC ($\mu m^2$) | 3625.71 | 23.49 | 0.875 (0.756) | 3567.09 | 140.71 | 0.851 (0.714) |
| SC meridional diameter ($\mu m$) | 258.87 | 13.80 | 0.895 (0.794) | 252.05 | 27.43 | 0.814 (0.649) |
| TIA ($\mu m$) | 59.51 | 3.67 | 0.832 (0.680) | 59.63 | 3.43 | 0.804 (0.632) |
| TM thickness ($\mu m$) | 105.05 | 4.06 | 0.912 (0.825) | 110.56 | 6.9 | 0.946 (0.891) |
| Iris thickness ($\mu m$) | 201.59 | 1.23 | 0.842 (0.698) | 199.53 | 2.89 | 0.872 (0.752) |
| Ciliary process length ($\mu m$) | 1684.50 | 103.93 | 0.842 (0.698) | 1679.04 | 114.86 | 0.821 (0.661) |
| Corneal limbus thickness ($\mu m$) | 712.20 | 17.40 | 0.846 (0.705) | 709.10 | 12.93 | 0.841 (0.696) |

SC: Schlemm's canal; TIA: Trabecular-iris angle; TM, trabecular meshwork; ICC: Intraclass correlation coefficient; CI: confidence interval.

## Quantitative anterior chamber angle parameters

Quantitative parameters of the anterior chamber angle for groups 1 and 2 are shown in Table 3. Notably, CSA of SC, SC meridional diameter, iris thickness, and corneal limbus thickness were all significantly smaller in eyes with PCG (Group 1) compared to unaffected eyes (Group 2, all P<0.05). While TIA and ciliary process length in Group 2 was smaller than those in Group 1 (both P<0.05). There were no statistical differences between groups in TM thickness (P = 0.072). IOP was not related to the largest area of SC in either group by Generalized Estimating Equations (Group 1: P = 0.510; Group 2: P = 0.455) (Fig 3).

## Anterior chamber angle parameters and age in PCG eyes

The largest CSA of SC, TM thickness, iris thickness and ciliary process length were significantly correlated to age in 51 PCG eyes. The slopes and intercepts for the regression analysis, with correlation coefficients with p values, $R^2$ values and regression quotation are shown in Fig 4. IOP was not correlated with age in PCG eyes (P = 0.234).

## Discussion

Precise *in vivo* measurements of the developing anterior segment are essential in understanding the pathogenesis of congenital glaucoma [6–8,14]. It also provides useful information in planning glaucoma surgery [13,15]. Measurements of the TM and SC *in vivo* using spectral-domain optical coherence tomography (SD-OCT) or swept-source optical coherence tomography (SS-OCT) have been reported in older children [19–21]; however, this modality is not

**Table 3. Comparison of quantitative parameters of the anterior chamber angle between groups.**

| Group | Group 1 | Group 2 | P |
|---|---|---|---|
| Largest CSA of Schlemm's canal ($\mu m^2$, mean±SD) | 3363.91±1082.98 | 5130.66±1231.90 | <0.001* |
| SC meridional diameter ($\mu m$, mean±SD) | 257.70±66.70 | 335.09±104.76 | 0.009* |
| TIA ($\mu m$, mean±SD) | 64.52±15.28 | 48.02±15.62 | 0.002* |
| TM thickness ($\mu m$, mean±SD) | 111.52±40.79 | 95.37±25.72 | 0.072 |
| Iris thickness ($\mu m$, mean±SD) | 188.10±40.62 | 235.25±65.34 | 0.002* |
| Ciliary process length ($\mu m$, mean±SD) | 1498.88±300.02 | 1278.37±130.28 | 0.018 |
| Corneal limbus thickness ($\mu m$, mean±SD) | 702.73±100.37 | 771.62±78.59 | 0.030* |

* Statistical significance; Group 1, primary congenital glaucoma eyes; Group 2, unaffected contralateral eyes. CSA, cross-sectional area; SC, Schlemm's canal. TIA, trabecular-iris angle; TM, trabecular meshwork.

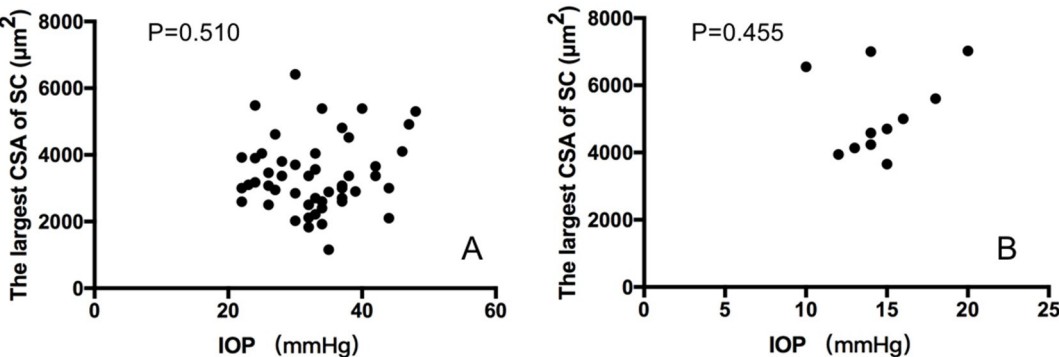

**Fig 3.** Generalized Estimating Equations analysis of intra-ocular pressure (IOP) and the largest cross-sectional area (CSA) of Schlemm's canal (SC) in 51 primary congenital glaucoma (PCG) eyes (A) and 11 contralateral unaffected eyes (B), and IOP was not related to the largest CSA of SC in either group.

possible in infants and younger children due to the amount of cooperation needed to complete the test, which can only be performed in a seated, awake child. The iUltrasound imaging system is suitable for examination in infants and children under general anesthesia in the supine position and both TM and SC can be clearly identified [10–12]. To our knowledge, only one prior study has measured SC diameter in PCG eyes using this modality [13]. Our study is the

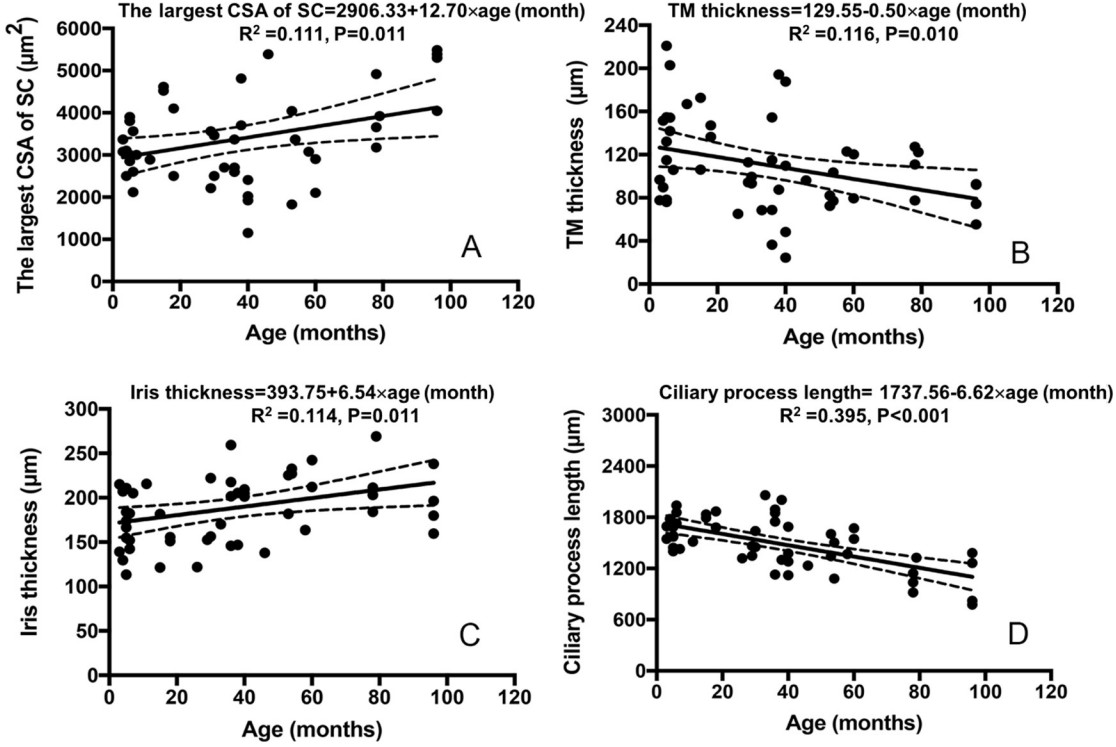

**Fig 4. Generalized Estimating Equations analysis of age and anterior chamber angle parameters in 51 PCG eyes.** (A) The largest cross-sectional area (CSA) of Schlemm's canal (SC) in relation to age (months) with mean and 95% prediction limits. (B) Iris thickness in relation to age (months) with mean and 95% prediction limits. (C) TM thickness in relation to age (months) with mean and 95% prediction limits. (D) Ciliary process length in relation to age (months) with mean and 95% prediction limits. The slopes and intercepts for the regression analysis, with correlation coefficients with p values, $R^2$ values and regression quotation are shown.

first to provide an in vivo quantitative assessment of the TM and SC, and explore their relationship to age in patients with PCG.

Previous study of Schlemm's canal in younger patients is notably limited. Using the 80-MHz UBM, Tandon and colleagues found Schlemm's canal could be identified in 62.5% of patients with PCG (age range: 3 days to 3 years) and in all 19 normal eyes (age range: 7 weeks-17 years) [13]. Since we excluded 9 eyes with no identified SC in any quadrants, we were able to observe Schlemm's canal in 73.1% of all quadrants with PCG and in 100% of normal eyes. Lack of proper development of SC has been postulated to play a role in the pathogenesis of primary congenital glaucoma [22–25], and its absence has been associated with a severe form of goniodysgenesis [26]. Inability to detect SC might be a result of segmental outflow as previously described in healthy adult eyes [27–29] or may affected by variation in IOP in patients with primary open-angle glaucoma [30]. For these reasons, we cannot fully extrapolate our finding of lower visibility of SC to the pathogenesis of PCG yet. However, the lack of SC was reported to be associated with poor prognosis of angle surgeries in PCG [4,26] identification of SC under 80-MHz UBM might be useful to plan angle surgery in these eyes.

Most current quantitative measurements of SC are confined to histologic study *in vitro* and have reported SC diameter from 180 to 250 μm in non-glaucomatous pediatric eyes and 92 to 250 μm in pediatric eyes with glaucoma [2]. The SC meridional diameter in our study was slightly larger in both groups compared to histological studies though smaller in eyes with PCG than with normal eyes (257.70±66.70 μm in PCG; 335.09±104.76 in normal eyes, P = 0.009). This could represent a difference in *in vivo* versus *in vitro* measurement as fixation and preparation methods may significantly alter the size and shape of SC from natural settings. Our investigation of Schlemm's canal *in vivo* may better reflect the canal's physiologic and functional dimensions. Moreover, SC diameter may dynamically vary depending on the amount of aqueous fluid. In this study, we chose the largest CSA of SC among 4 quadrants, which may lead to measure the largest possible diameter of SC. Using the 80 MHz anterior segment ultrasound, only Tandon et al. reported the single meridional canal diameter in pediatric nonglaucomatous eyes (142 ± 33.2 μm) was larger than average meridional canal diameter of 4 quadrants in pediatric glaucomatous eyes (64.9 ± 10.90 μm; *P* = 0.007), but they only recruited 10 subjects in each group and 50% glaucoma eyes failed to identify the SC [13].

Similarly, the largest CSA of SC in PCG eyes was significantly smaller than that of unaffected eyes (3363.91±1082.98 μm$^2$ compared to 5130.66±1231.90μm$^2$, P<0.001). No previous studies have reported the CSA of SC in PCG eyes, but in healthy adults using SD-OCT and SS-OCT, the SC CSA varies rapidly within short distances along its arc, ranging from 4064 ±1308 μm$^2$ to 13991±1357 μm$^2$ and may decrease with age [19,31–33]. Eyes with primary open angle glaucoma have a reduced CSA compared with normal healthy controls [33], and reduced SC size may be associated with elevated IOP because the size of SC is related to outflow facility [33,34].

The smaller dimensions of Schlemm's canal in children, as seen in our study, are well known [35,36]. We detected a significant increase with age in PCG eyes, and significantly smaller area in eyes with PCG compared to unaffected eyes. However, we didn't find a correlation between SC CSA and IOP in either group. We speculate that the smaller largest-measured SC area we observed in PCG eyes compared with unaffected eyes may not be a result of elevated IOP but related to disease pathogenesis. Also, the elevation of IOP may related to the degree of the maldeveloped SC, but we didn't have enough data in this study to prove this. We only performed UBM at 4 o'clock, not 360-degree. The future development of circumference SC image would help solve this question.

The trabecular beams in PCG have been described using microscopy as thickened and compacted, increasing outflow resistance [25,37]. In our study, trabecular meshwork thickness in

eyes with PCG was thicker than unaffected eyes, but this difference was not statistically significant. This is consistent with prior studies showing trabecular meshwork thickness of 107 ±51 μm from trabeculotomy specimens of patients with infantile glaucoma (average age 8 years) [3]. Similar to ours results, that study shows a decrease in trabecular meshwork thickness with age in patients with glaucoma in contrast to the increase in trabecular meshwork thickness with age reported in healthy individuals [19]. However, the presence of an anterior chamber membrane may have affected our ability to accurately measure trabecular meshwork thickness. Twenty seven percent of glaucoma patients had identifiable abnormal tissue in our cohort. The rate of identifying the anterior chamber membrane varies in the literature, ranging from 100% to 12%.[5] One study identifying abnormal tissue in 100% of eyes using SD-OCT suffered from poor inter-grader agreement (κ = 0.61, P < 0.005) [21]. The lower rate of membrane identification in this study may be due to our strict criteria requiring agreement between two glaucoma specialists where equivocal cases were excluded. Further studies comparing histological and UBM results may better determine if high-resolution ultrasound is a precise method to measure trabecular meshwork thickness *in vivo*.

The trabecular-iris angle was larger in eyes with primary congenital glaucoma compared to unaffected eyes, and iris thickness was thinner and ciliary process length was longer in PCG eyes compared with unaffected eyes, consistent with prior studies [5,6,8,38]. Although ridges or crypts of the iris surface may affect the measurement of iris thickness, loss of normal iris configuration in PCG similar to our findings has been reported [5]. Iris thickness increases with aging, while ciliary process length decreases with aging. Although it might denote that the thinning of the iris and ill-defined ciliary process could be the result of progressive stretching of the globe in these eyes, it could also indicate dysplasia of the anterior chamber angle in PCG with concomitant abnormal iris and ciliary process, which could still develop with age in PCG eyes.

We detected that the corneal limbus thickness was thinner in PCG eyes compared to unaffected eyes. Both thinner [39–41] and thicker [42,43] central cornea thickness (CCT) have both been reported in PCG patients, explained by corneal stretching and/or scarring for thinner CCT or an inherent component of the pathophysiology related to racial and genetic background or edema for thicker CCT. Since scarring and edema rarely affect the limbus in PCG eyes, we believe that the thinner corneal limbus thickness may better illustrate the corneal stretching due to enlargement of eye ball under high IOP.

The current study reports a bilateral PCG incidence of about 72%, consistent with rates in the literature: 60%-99.3% [44]. This higher incidence of bilaterality in other reports may be related to disease severity or underlying genetic abnormality in different ethnic groups. Although we did find the anterior segment structures were significantly different between affected PCG eyes and unaffected eyes, we still cannot determine if severe goniodysgenesis is the cause or the result of the disease. Differences between affected and unaffected eyes in patients with unilateral PCG patients shed insight into this issue; however, there were only 6 unilateral PCG patients with both affected and unaffected eyes recruited in this study, so further study is warranted to explore this relationship.

The present study has certain limitations. Firstly, observer and measurement errors might exist in that all measurements were taken by a single masked observer, though we attempted to minimize this effect by obtaining a consensus of two observers on the identification of key angle structures. Analysis of the intraobserver and interobserver reproducibility was high. Secondly, our study has a small sample size, but is acceptable given the incidence of primary congenital glaucoma. Thirdly, with regards to measurement of the largest cross-sectional area of Schlemm's canal, it is possible that this varies from quadrant to quadrant with natural variation at different anatomic locations with the largest cross-sectional area not comprehensively

reflecting the anatomy of the whole eye, and the inability to identify SC in the superior quadrant alone could be due to imaging technique for those enlarged PCG eyes and not a consistent anatomical finding. Fourthly, PCG is a genetic disorder and it is likely that the "unaffected" eye is not truly normal [44]. Further studies should address the differences of anterior chamber angle structures between unaffected eyes of PCG patients and normal age-matched eyes. Finally, the cross-sectional nature of our study did not allow us to comprehensively elucidate age-related changes in the development of anterior chamber angle in PCG eyes, and longitudinal studies are warranted to further explore this relationship.

## Conclusions

Notwithstanding the limitations, we believe that this study is the first of its kind to provide quantitative information about anterior segment morphology and its relationship with age *in vivo* using high resolution UBM in eyes with PCG. The anatomical information gleaned in the study demonstrates the usefulness of high-frequency UBM in understanding the angle dysgenesis, and it might also be useful in planning anterior segment surgery in these eyes based on the development of the anterior chamber angle.

## Supporting information

**S1 Data.**
(SAV)

**S2 Data.**
(SAV)

## Author Contributions

**Conceptualization:** Yan Shi, Huaizhou Wang, Ningli Wang.

**Data curation:** Yan Shi, Chen Xin, Man Hu, Kai Cao.

**Formal analysis:** Yan Shi, Ying Han, Chen Xin.

**Funding acquisition:** Yan Shi.

**Investigation:** Yan Shi, Huaizhou Wang, Ningli Wang.

**Supervision:** Huaizhou Wang, Ningli Wang.

**Writing – original draft:** Yan Shi.

**Writing – review & editing:** Ying Han, Julius Oatts.

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
