## [Decision Letter · Decision Letter 0]

16 Sep 2019

PONE-D-19-17802

Disease-related and age-related changes of anterior chamber angle structures in patients with primary congenital glaucoma: An in vivo high-frequency ultrasound biomicroscopy-based study

PLOS ONE

Dear M.D., Ph.D. Wang,

Thank you for submitting your manuscript to PLOS ONE. After careful consideration, we feel that it has merit but does not fully meet PLOS ONE’s publication criteria as it currently stands. Therefore, we invite you to submit a revised version of the manuscript that addresses all the points raised by the reviewers and the comments below..

The study investigates features of primary congenital glaucoma (PGC) using ultrasound biomicroscopy.

1. The small sample size for unaffected eyes (controls) may impact significance of the findings. Please justify.

2. The controls were all contralateral eyes of unilateral cases of PGC. One study recently suggested that the normal eyes in unilateral PGC may not be anatomically normal (Bayoumi 2017) - can the authors please comment, and also indicate if they have  compared their findings to data for patients with both eyes unaffected. 

3. The issue of unilateral versus bilateral PGC is interesting and should be commented upon further. What is the prevalence of each condition in China and across other countries?  Is there any data on the underlying causes of PGC for the study participants? Have genetic mutations for example, been investigated? 

4. The methods used to assess the anterior chamber angle characteristics are unclear. Please revise.

5. The age-related effects for PGC eyes are presented in Figure 3, but there is limited discussion as to what these graphs mean.. How does this compare to normal unaffected eyes (noting very small sample size) (both contralateral eyes, and also both eyes unaffected).

We would appreciate receiving your revised manuscript by Oct 31 2019 11:59PM. To enhance the reproducibility of your results, we recommend that if applicable you deposit your laboratory protocols in protocols.io, where a protocol can be assigned its own identifier (DOI) such that it can be cited independently in the future. For instructions see: http://journals.plos.org/plosone/s/submission-guidelines#loc-laboratory-protocols

We look forward to receiving your revised manuscript.

Kind regards,

Michele Madigan

Academic Editor

PLOS ONE

Journal Requirements:

Reviewers' comments:

Reviewer's Responses to Questions

**Comments to the Author**

1. Is the manuscript technically sound, and do the data support the conclusions?

Reviewer #1: Yes

Reviewer #2: Partly

2. Has the statistical analysis been performed appropriately and rigorously? 

Reviewer #1: Yes

Reviewer #2: No

3. Have the authors made all data underlying the findings in their manuscript fully available?

Reviewer #1: Yes

Reviewer #2: No

4. Is the manuscript presented in an intelligible fashion and written in standard English?

Reviewer #1: Yes

Reviewer #2: No

5. Review Comments to the Author

Reviewer #1: Please see comments to be addressed below. In addition to the below comments the only other concern with regard to statistical analysis is the relative small sample size of control "normal" eyes compared to affected eyes. If possible, someone more specialised in statistical analysis might be useful to check the validity of the analysis.

Line 45: Last sentence in conclusion should also state “and changes with age”

Line 48: The sentence “Primary congenital glaucoma is the leading cause of blindness worldwide” is incorrect.

PCG only accounts for 3-5% of blindness in children worldwide. A greater number are affected in developing countries. It is the leading cause of blindness in Saudi Arabia and the Rom population of Slovakia. It is however a very rare condition that is difficult to treat and detect and can often be missed which is why this study is interesting.

Line 84: Sclera spur should be changed to scleral spur

Line 91: For imaging of the anterior chamber angle were the scans acquired by a technician and then interpreted by the two investigators? And if so, was it the same person that did all the scans?

Line 161: 51 patients with 75 eyes were underwent the UBM measurement (remove were)

For statistics:

1. Comparing 51 affected eyes to 11 “normal/control” eyes not sure how reliable it is to do hypothesis testing here as there is a small comparative sample size

For results:

1. Table 1: Group 1 column, under “observable SC proportion”

a. Total (n, %). 152/208 (*where does the value of 208 come from? If the SC is observed in 4 quadrants per eye should it not be 204?)

b. For each of the remaining regions (superior, nasal, inferior and temporal) should the fraction be /51 instead of 52?

Line 319: should read “which was consistent with the results in other studies”

Line 334: In addition of the study limitations mentioned I wonder if the following two points should also be included:

1. Small comparison sample size. Only 11 contralateral unaffected eyes compared to 51 affected eyes, this does make comparison between groups difficult

2. Is the supposedly unaffected eye in unilateral disease actually normal? PCG is a genetic disorder and it is likely that the “unaffected” eye is not truly normal.

It would be interesting to know if the eyes that were excluded with complete absence of SC had angle surgery or went straight to either a trabeculectomy or glaucoma drainage device? Presumably absence of SC would mean that angle surgery including trabeculotomy would be contraindicated. In your conclusion an additional point to be mentioned would be that UBM could be used to guide surgical options.

Reviewer #2: This study on the application of UBM to primary congenital glaucoma is an interesting one. This work has several key messages that would contribute significantly to the field. My main comment is regarding the clarity of the methods and the unfortunate dilution of some of the key messages of the paper. I have some other comments for the authors to address about their work.

Introduction:

Line 52-53: The introduction clearly outlines a need to understand the mechanism of PCG. To be clear though, the use of UBM as an instrument would provide insight into the ocular structures involved in primary congenital glaucoma, though the mechanism by which primary congenital glaucoma occurs is something that would probably remain elusive, e.g. a genetic cause. It may be useful for the authors to clarify this.

Line 61-62: It would be useful to be more critical in the description of UBM use. In adult glaucoma, it is more commonly used to verify the presence of plateau iris syndrome for example rather than routine use, and importantly it contrasts with the greater utility of anterior segment OCT in general practice. Part of the issue here is the statement regarding "precise measurements", which is simply not comparable to high resolution instruments such as AS OCT (e.g. Liebmann & Ritch 1996). Although it is stated later one, it is worthwhile clarifying that there is an improvement in axial resolution compared to 50 MHz UBM.

Lines 66-67: there is a disconnect between this sentence and the above paragraphs as the authors make the leap of logic from understanding the mechanism of PCG to "pathogenesis and management".

Methods:

- Line 82: "corresponding"

- Lines 98-99: it is not clear if the ultrasound recordings were in video form or in image frame form when it is expressed in "20 ultrasound recordings". It is worthwhile clarifying here.

- Lines 108-111: Could the authors comment on the issue identified by Tandon et al 2017 J AAPOS and Yan et al 2016 PLOS who found that 50% of the time Schlemm's canal could not be identified in PCG?

- The methods in lines 114 onwards are a little bit confusing an lack sufficient detail. For example, the cross-sectional area "taken at four different positions" is not clear. Is it four per quadrant or four in total? I'm not sure how the "largest of which was used for analysis to account for any variability" would reduce variability and not in fact introduce a systematic bias? The trabecular-iris angle is an interesting choice of parameter. What happens when there are irregularities in the anterior iris surface, for example, the presence of ridges or crypts? It would be more informative to state at which distance, similar to the way that AOD is measured for example, the TIA was taken. Corneal limbus thickness is poorly defined: is the the shortest distance or the perpendicular to the limbus tangent? Line 131 "average values" -- of what?

- Statistical analysis: intraobserver variance was only measured in 18 eyes... at this point of the manuscript, it is not clear how significant this number is relative to the proportion of the sample size. It is more informative to state, e.g. 20% of the eyes were randomly selected for re-evaluation. I note that this was only for a single observer - was this just for the measurements and not the delineation? What if there were issues with landmarks? The fidelity of the measurements is highly dependent upon accurate delineation of landmarks and so that is also important to assess for inter- and intraobserver variability.

Results:

- Table 1: how come total is out 208 for Group 1 when there are 51 eyes? Should it not be 204? Also, would it not be more informative to have proportion in terms of "number of eyes with the largest CSA of SC in each quadrants" as well? Otherwise, it is currently confusing and without context.

- Line 186: this is a very key and interesting finding but it is lost amidst this paragraph and the use of abbreviations. My suggestion is to rename "group 1" and "group 2" as they are currently meaningless and consider instead "Bilateral disease" and "Unilateral disease" instead.

- Given the large number of parameters being examined in this study, I would suggest a Table in the methods to list out the relevant parameters as well.

- It might be worthwhile combining Figure 3 and Table 4 which state the same thing, with the regression equations being put in as insets.

- The relationship between IOP and other parameters was sparingly mentioned in the Results. It may be worthwhile showing these figures so that the reader can contrast these results with the work of Yan et al 2016 PLOS

Discussion/conclusions:

- Lines 238-242 should really be put in the introduction to highlight the importance of UBM.

- Lines 243-246: I'm not sure if this claim is fully supported. There are other papers in the literature that report on quantitative assessment of the anterior chamber structures in PCG (e.g. Gupta et al 2007 J AAPOS, Hussein et al 2014 Clin Ophthalmol)

- Lines 250 onwards: what is an interesting question here is whether other meaningful parameters can be assessed in patients in whom SC cannot be visualised. This is worth discussing and even reporting if the data is available. Given that such a large proportion of patients fit into this criteria of SC non-visibility, it would be highly informative and contributory to the literature.

- The discussion is generally very long and perhaps unnecessarily so given the length of the results. I refer specifically to the paragraphs between lines 287-324.

- What I feel is a very interesting result in the unilateral versus bilateral comparison group was not really discussed.

- Conclusions (line 352) the idea of age needs to be mentioned throughout the manuscript if this claim is to be made. Right now, it is relatively sparse.

Miscellaneous comments:

- The overall writing is generally clear. There are minor grammatical errors that should be carefully reviewed.

- Data availability: I don't see where the data is/will be made available at this stage of the review process.

6. PLOS authors have the option to publish the peer review history of their article (what does this mean?). If published, this will include your full peer review and any attached files.

Reviewer #1: No

Reviewer #2: No

---

## [Author Response · Author response to Decision Letter 0]

20 Oct 2019

The study investigates features of primary congenital glaucoma (PGC) using ultrasound biomicroscopy.

1. The small sample size for unaffected eyes (controls) may impact significance of the findings. Please justify.

 Thank you for the comment. We realize our sample size is small due to the low incidence of primary congenital glaucoma. We have included a detailed statistical power calculation as it relates to our sample size in our response to Reviewer 1 below.

2. The controls were all contralateral eyes of unilateral cases of PGC. One study recently suggested that the normal eyes in unilateral PGC may not be anatomically normal (Bayoumi 2017) - can the authors please comment, and also indicate if they have compared their findings to data for patients with both eyes unaffected. 

 We agree completely that unaffected eyes of children with unilateral PCG are not totally normal. In future studies, we plan to address the differences of anterior chamber angle structures between unaffected eyes of PCG patients and normal age-matched eyes. Given the small number of unaffected eyes in patients with unilateral PCG in this study, we have further discussed this in the limitations portion of our manuscript. (Lines 359 to 362)

3. The issue of unilateral versus bilateral PGC is interesting and should be commented upon further. What is the prevalence of each condition in China and across other countries?  Is there any data on the underlying causes of PGC for the study participants? Have genetic mutations for example, been investigated? 

 The current study reports an incidence of bilateral PCG of 72%, in line with prior published reports: 60%-99.3%. We included this information in our revised manuscript. (Lines 338 to 347) While outside the scope of this paper, we are currently researching the genetic background of unilateral and bilateral PCG.

4. The methods used to assess the anterior chamber angle characteristics are unclear. Please revise.

 We have revised them according to the reviewers’ comments.

5. The age-related effects for PGC eyes are presented in Figure 3, but there is limited discussion as to what these graphs mean. How does this compare to normal unaffected eyes (noting very small sample size) (both contralateral eyes, and also both eyes unaffected).

 We had discussed the age-related effects for PCG eyes (Line 290, Line 295, Lines 305 to 308, Lines 324 to 329, Lines 362 to 365). Unfortunately, there were so limited previously published data about this issue. Moreover, the cross-sectional nature of our study did not allow us to comprehensively elucidate age-related changes in the development of anterior chamber angle in PCG eyes, and longitudinal studies are warranted to further explore this relationship. We plan to evaluate this in future studies, and even to compare our findings to data for patients with both normal eyes, as the unaffected eyes in unilateral PCG may not be anatomically normal as discussed above (Bayoumi 2017).

Reviewers' comments:

Reviewer #1: Please see comments to be addressed below. In addition to the below comments the only other concern with regard to statistical analysis is the relative small sample size of control "normal" eyes compared to affected eyes. If possible, someone more specialised in statistical analysis might be useful to check the validity of the analysis.

 Thank you for your advice. We had a statistician check the validity of the analysis and have included additional details below.

Line 45: Last sentence in conclusion should also state “and changes with age”

 This has been corrected. (Line 44)

Line 48: The sentence “Primary congenital glaucoma is the leading cause of blindness worldwide” is incorrect. PCG only accounts for 3-5% of blindness in children worldwide. A greater number are affected in developing countries. It is the leading cause of blindness in Saudi Arabia and the Rom population of Slovakia. It is however a very rare condition that is difficult to treat and detect and can often be missed which is why this study is interesting.

 This has been corrected. (Line 47)

Line 84: Sclera spur should be changed to scleral spur

 This has been corrected. (Line 85)

Line 91: For imaging of the anterior chamber angle were the scans acquired by a technician and then interpreted by the two investigators? And if so, was it the same person that did all the scans?

 All the scans were acquired by the same investigator (YS) and then interpreted by two investigators (YS and CX). We have added this information to the manuscript. (Line 94)

Line 161: 51 patients with 75 eyes were underwent the UBM measurement (remove were)

 We have corrected this sentence. (Line 167)

For statistics:

1. Comparing 51 affected eyes to 11 “normal/control” eyes not sure how reliable it is to do hypothesis testing here as there is a small comparative sample size

 Literature on imbalanced designs indicates that a 1:1 ratio may not be optimal in terms of statistical precision or costs.1,2 This is especially true for situations where the exposure is rare and for stronger relationships between the exposure and the outcome under study.3 Since the PCG is commonly bilateral, the unequal sample size of affected eyes and unaffected control eyes is inevitable. Generally, the formula for calculating sample size of an unmatched case–control study is

n_1=(〖〖(Z〗_(1-α/2)+Z_(1-β))〗^2×(σ_1^2+σ_2^2)×(1+1/K))/δ^2 

n_1: sample size for group 1. : sample size for group 2.

K: the ratio of cases in two groups

σ_1: the standard deviation of group 1

σ_2: the standard deviation of group 1

δ: the mean difference between groups

The main outcome of this study is the largest CSA of Schlemm’s canal, which were 3363.91±1082.98 μm2 in group 1 and 5130.66±1231.90 μm2 in group 2. The ratio of affected/unaffected eyes was 4.6. Using a two-sided test with significance of 0.05 and a power of 80%, the minimum required sample size of affected eye was 16.5 and 3.6 unaffected eyes. Since we included both eyes of patients with bilateral disease, we tripled the minimum sample size in each group, comparing 51 affected eyes to 11 unaffected eyes.

1. Nam JM. Optimum sample sizes for the comparison of the control and treatment. Biometrics. 1973;29(1):101-8.

2. Liu X. Statistical power and optimum sample allocation ratio for treatment and control having unequal costs per unit of randomization. J Edu Behav Stats. 2003;28(3):231-48.

3. Groenwold R H H , Smeden M V . Efficient Sampling in Unmatched Case-Control Studies When the Total Number of Cases and Controls Is Fixed.[J]. Epidemiology, 2017, 28(6):834.

For results:

1. Table 1: Group 1 column, under “observable SC proportion”

a. Total (n, %). 152/208 (*where does the value of 208 come from? If the SC is observed in 4 quadrants per eye should it not be 204?)

b. For each of the remaining regions (superior, nasal, inferior and temporal) should the fraction be /51 instead of 52?

 Thank you for finding these errors – we have corrected them.

Line 319: should read “which was consistent with the results in other studies”

 This has been corrected. (Line 321)

Line 334: In addition of the study limitations mentioned I wonder if the following two points should also be included:

1. Small comparison sample size. Only 11 contralateral unaffected eyes compared to 51 affected eyes, this does make comparison between groups difficult

 The issue of sample size was addressed above. The small number of patients with unilateral PCG patients created the disparity between the number of affected and unaffected eyes. We added this to our discussion as well. (Lines 344 to 347)

2. Is the supposedly unaffected eye in unilateral disease actually normal? PCG is a genetic disorder and it is likely that the “unaffected” eye is not truly normal.

 As above, we have added this to the study limitations. (Lines 359 to 362)

It would be interesting to know if the eyes that were excluded with complete absence of SC had angle surgery or went straight to either a trabeculectomy or glaucoma drainage device? Presumably absence of SC would mean that angle surgery including trabeculotomy would be contraindicated. In your conclusion an additional point to be mentioned would be that UBM could be used to guide surgical options.

We do currently use UBM to guide surgical options, especially for consideration of microcatheter-assisted trabeculotomy (MAT). We have previously reported on the correlation of goniodysgenesis as evaluated by UBM and outcomes of MAT in eyes with PCG.1 

 All PCG patients in this study underwent the MAT. However, in 9 patients with complete absence of SC on UBM, we can still catheterize Schlemm’s canal 360-degree (2 eyes) or partially (3 eyes). Schlemm’s canal could not be identified in 4 eyes during the surgery, and then they underwent the traditional trabeculotomy using Harm’s trabeculotome. Predicting the degree of successful catheterization based on UBM SC visualization can be challenging. We only tested four positions of the SC using UBM, while a lack of SC visibility in these four positions may result from segmental outflow of SC, as detected in healthy eyes2-4 or may be affected by the IOP, as found in primary open-angle glaucoma.5 So diminished visualization of SC under UBM may suggest a decreased possibility of full 360° catheterization, but further studies were warranted. We mention the role of UBM in surgery in our discussion. (Lines 265 to 268)

1. Shi Y, Wang HZ, Han Y, Cao K, Vu V, Hu M, et al. Correlation between trabeculodysgenesis assessed by ultrasound biomicroscopy and surgical outcomes in primary congenital glaucoma. Am J Ophthalmol. 2018;196:57-64.

2. Keller KE, Bradley JM, Vranka JA, Acott TS. Segmental versican expression in the trabecular meshwork and involvement in outflow facility. Invest Ophthalmol Vis Sci. 2011 Jul 7;52(8):5049-57. 

3. Yang CY, Liu Y, Lu Z, Ren R, Gong H. Effects of Y27632 on aqueous humor outflow facility with changes in hydrodynamic pattern and morphology in human eyes. Invest Ophthalmol Vis Sci. 2013 Aug 28;54(8):5859-70.

4. Hann CR, Fautsch MP. Preferential fluid flow in the human trabecular meshwork near collector channels. Invest Ophthalmol Vis Sci. 2009 Apr;50(4):1692-7.

5. Tandon A, Watson C, Ayyala R. Ultrasound biomicroscopy measurement of Schlemm's canal in pediatric patients with and without glaucoma. J AAPOS. 2017;21(3):234-237.

Reviewer #2: This study on the application of UBM to primary congenital glaucoma is an interesting one. This work has several key messages that would contribute significantly to the field. My main comment is regarding the clarity of the methods and the unfortunate dilution of some of the key messages of the paper. I have some other comments for the authors to address about their work.

Introduction:

Line 52-53: The introduction clearly outlines a need to understand the mechanism of PCG. To be clear though, the use of UBM as an instrument would provide insight into the ocular structures involved in primary congenital glaucoma, though the mechanism by which primary congenital glaucoma occurs is something that would probably remain elusive, e.g. a genetic cause. It may be useful for the authors to clarify this.

 Thank you for this insightful comment. We have updated our language to reflect this important point. (Lines 50 to 52) 

Line 61-62: It would be useful to be more critical in the description of UBM use. In adult glaucoma, it is more commonly used to verify the presence of plateau iris syndrome for example rather than routine use, and importantly it contrasts with the greater utility of anterior segment OCT in general practice. Part of the issue here is the statement regarding "precise measurements", which is simply not comparable to high resolution instruments such as AS OCT (e.g. Liebmann & Ritch 1996). Although it is stated later one, it is worthwhile clarifying that there is an improvement in axial resolution compared to 50 MHz UBM.

 Thank you for your reminder. We have updated our manuscript to include these points. (Lines 57 to 58, Lines 62 to 63, 65)

Lines 66-67: there is a disconnect between this sentence and the above paragraphs as the authors make the leap of logic from understanding the mechanism of PCG to "pathogenesis and management".

 We have revised this. (Line 68)

Methods:

- Line 82: "corresponding"

 We have corrected it. (Line 83)

- Lines 98-99: it is not clear if the ultrasound recordings were in video form or in image frame form when it is expressed in "20 ultrasound recordings". It is worthwhile clarifying here.

 These were images – we have clarified this. (Line 100)

- Lines 108-111: Could the authors comment on the issue identified by Tandon et al 2017 J AAPOS and Yan et al 2016 PLOS who found that 50% of the time Schlemm's canal could not be identified in PCG?

 We address identification of Schlemm’s canal in the discussion section. (Lines 254 to 268)

- The methods in lines 114 onwards are a little bit confusing an lack sufficient detail. For example, the cross-sectional area "taken at four different positions" is not clear. Is it four per quadrant or four in total? 

 We have updated the methods to provide additional clarity. (Lines 118 to 119)

I'm not sure how the "largest of which was used for analysis to account for any variability" would reduce variability and not in fact introduce a systematic bias? 

 We have included a discussion of variation of the cross-sectional area of Schlemm’s canal in the discussion section. We agree with you that the largest cross-sectional area may not be representative of the anatomy of the whole eye, but it did serve as a standardized measurement for the purposes of our study. (Lines 353 to 359)

The trabecular-iris angle is an interesting choice of parameter. What happens when there are irregularities in the anterior iris surface, for example, the presence of ridges or crypts?

 Loss of normal iris configuration has been reported in PCG eyes1, though we did not observe any ridges or crypts of the iris surface in our cohort of PCG eyes. We have added a comment to address this in the discussion section. (Lines 322 to 324)

1. Hussain T, Shalaby S, Elbakary MA, Elseht R, Gad R. Ultrasound biomicroscopy as a diagnostic tool in infants with primary congenital glaucoma. Clin Ophthalmol. 2014:1725-1730.

It would be more informative to state at which distance, similar to the way that AOD is measured for example, the TIA was taken. 

 This was measured 500 μm from the scleral spur. We have updated the methods with a detailed description. (Lines 128 to 130)

Corneal limbus thickness is poorly defined: is the shortest distance or the perpendicular to the limbus tangent?

 This was measured perpendicular to the limbus tangent. Methods have been updated to reflect this. (Lines 133 to 134)

Line 131 "average values" -- of what?

 This has been updated. (Line 136)

- Statistical analysis: intraobserver variance was only measured in 18 eyes... at this point of the manuscript, it is not clear how significant this number is relative to the proportion of the sample size. It is more informative to state, e.g. 20% of the eyes were randomly selected for re-evaluation. I note that this was only for a single observer - was this just for the measurements and not the delineation? What if there were issues with landmarks? The fidelity of the measurements is highly dependent upon accurate delineation of landmarks and so that is also important to assess for inter- and intraobserver variability.

 We have re-evaluated the intraobserver and interobserver variances using a randomly selected 50% of the eyes. As the accurate delineation of landmarks was quite important for our measurement, the landmarks (SC and SS) were delineated by two observers together to ensure a similar understanding of the anatomical landmarks. When the observers disagreed on the delineation of landmarks, a mutual conclusion was reached after discussion. For this reason, we didn’t analyze the interobserver variability, since the landmarks was agreed up by the two observers. Our result showed a good interobserver and intraobserver reproducibility. (Table 2, Lines 160 to 164, Lines 351)

Results:

- Table 1: how come total is out 208 for Group 1 when there are 51 eyes? Should it not be 204? Also, would it not be more informative to have proportion in terms of "number of eyes with the largest CSA of SC in each quadrants" as well? Otherwise, it is currently confusing and without context.

 We have updated this table to incorporate these suggestions. (Table 1)

- Line 186: this is a very key and interesting finding but it is lost amidst this paragraph and the use of abbreviations. My suggestion is to rename "group 1" and "group 2" as they are currently meaningless and consider instead "Bilateral disease" and "Unilateral disease" instead.

 Thank you for this excellent suggestion – we have made the change to emphasize the importance of this finding. We have also updated our abstract to reflect this distinction. (Line 28, Lines 34 to 50, Line 192, Line 197)

- Given the large number of parameters being examined in this study, I would suggest a Table in the methods to list out the relevant parameters as well.

 Thank you for your suggestion. We agree that the large number of parameters is difficult to convey. Our hope is that our changes to Table 2 and 3 clarify this.

- It might be worthwhile combining Figure 3 and Table 4 which state the same thing, with the regression equations being put in as insets.

 Thank you for your suggestion. We have deleted Table 4 and put the regression equations in Figure 4.

- The relationship between IOP and other parameters was sparingly mentioned in the Results. It may be worthwhile showing these figures so that the reader can contrast these results with the work of Yan et al 2016 PLOS

 We have added this to Figure 3.

Discussion/conclusions:

- Lines 238-242 should really be put in the introduction to highlight the importance of UBM.

 We have highlighted this point in the introduction. (Lines 57 to 58)

- Lines 243-246: I'm not sure if this claim is fully supported. There are other papers in the literature that report on quantitative assessment of the anterior chamber structures in PCG (e.g. Gupta et al 2007 J AAPOS, Hussein et al 2014 Clin Ophthalmol)

 Our study is the first to provide an in vivo quantitative assessment of the TM and SC, and explore their relationship to age in patients with PCG. We have revised this sentence to reflect that. (Lines 251 to 253)

- Lines 250 onwards: what is an interesting question here is whether other meaningful parameters can be assessed in patients in whom SC cannot be visualised. This is worth discussing and even reporting if the data is available. Given that such a large proportion of patients fit into this criteria of SC non-visibility, it would be highly informative and contributory to the literature.

 Thank you for your suggestion. Our main outcomes were to measure SC and TM. When SC cannot be visualized, then SC and TM cannot be measured, which was why we excluded these cases.

- The discussion is generally very long and perhaps unnecessarily so given the length of the results. I refer specifically to the paragraphs between lines 287-324.

 We have condensed this section to make it more concise.

- What I feel is a very interesting result in the unilateral versus bilateral comparison group was not really discussed.

 We have updated our discussion to address this. (Lines 338 to 347)

- Conclusions (line 352) the idea of age needs to be mentioned throughout the manuscript if this claim is to be made. Right now, it is relatively sparse.

 Thank you for this feedback. The results section “Anterior Chamber Angle Parameters and Age in PCG Eyes” addresses this. (Line 226) Additionally, we have updated our discussion to include additional mention of this. (Line 290, Line 295, Lines 305 to 308, Lines 324 to 329, Lines 362 to 365)

Miscellaneous comments:

- The overall writing is generally clear. There are minor grammatical errors that should be carefully reviewed.

 Thank you for this feedback. We had our native English-speaking author (Julius Oatts) revised the manuscript.

- Data availability: I don't see where the data is/will be made available at this stage of the review process.

 We updated the data and submitted it as the supporting information according to the PLOS Data policy during the process of submitting our revised manuscript.

---

## [Decision Letter · Decision Letter 1]

9 Dec 2019

PONE-D-19-17802R1

Disease-related and age-related changes of anterior chamber angle structures in patients with primary congenital glaucoma: An in vivo high-frequency ultrasound biomicroscopy-based study

PLOS ONE

Dear M.D., Ph.D. Wang,

Thank you for submitting your manuscript to PLOS ONE. After careful consideration, we feel that it has merit but does not fully meet PLOS ONE’s publication criteria as it currently stands. Therefore, we invite you to submit a revised version of the manuscript that addresses the points raised during the review process.

Thank you for the detailed revision of the submitted manuscript.

Please note the following minor comments (and the Reviewer 2 comments below) and please address all points:

1. Please confirm that the iris thickness measurements were taken perpendicular to the posterior iris plane  and include this information in the Methods of the manuscript. 

2. Please indicate if trabecular iris space area (TISA) measurements (mm2) were included during the study - these are usually reported for UBM studies (either at 500um or 700um from the scleral spur). If the TISA measurements were taken, please include in the manuscript and discuss the outcomes.

3. Line 54: please amend to 'physiological.

4. Line 95: 'For those with unilateral disease, the unaffected contralateral eyes served as the control group (Group 2).'

5. Line 175: 'Interobserver agreement was calculated by comparing initial values of Observer 1 (YS)

to those of Observer 2 (CX).'

6. Line 382: 'Analysis of the intraobserver and interobserver reproducibility was high.'

7. Line 390: 'Fourthly,...'

We would appreciate receiving your revised manuscript by Jan 23 2020 11:59PM. To enhance the reproducibility of your results, we recommend that if applicable you deposit your laboratory protocols in protocols.io, where a protocol can be assigned its own identifier (DOI) such that it can be cited independently in the future. For instructions see: http://journals.plos.org/plosone/s/submission-guidelines#loc-laboratory-protocols

We look forward to receiving your revised manuscript.

Kind regards,

Michele Madigan

Academic Editor

PLOS ONE

Reviewers' comments:

Reviewer's Responses to Questions

**Comments to the Author**

1. If the authors have adequately addressed your comments raised in a previous round of review and you feel that this manuscript is now acceptable for publication, you may indicate that here to bypass the “Comments to the Author” section, enter your conflict of interest statement in the “Confidential to Editor” section, and submit your "Accept" recommendation.

Reviewer #2: (No Response)

2. Is the manuscript technically sound, and do the data support the conclusions?

Reviewer #2: Yes

3. Has the statistical analysis been performed appropriately and rigorously? 

Reviewer #2: Yes

4. Have the authors made all data underlying the findings in their manuscript fully available?

Reviewer #2: Yes

5. Is the manuscript presented in an intelligible fashion and written in standard English?

Reviewer #2: Yes

6. Review Comments to the Author

Reviewer #2: I thank the authors for their comprehensive response. I have a few minor comments below.

Line 127: I presume that the iris thickness was measured perpendicularly to the retroiridal iris plane?

My point about trabecular-iris angle was related more to why trabecular-iris space area was not used to account for iris surface anatomy. TISA is commonly used and I wonder why it has been omitted here.

Line 359: spelling error

With regard to intraocular pressure, how do the authors explain the big difference in pressure between Groups 1 and 2 even though there was no relationship found cross sectional area of Schlemm's canal (Figure 3)? Could it be related to the degree to which the canals were closed/narrow?

7. PLOS authors have the option to publish the peer review history of their article (what does this mean?). If published, this will include your full peer review and any attached files.

Reviewer #2: No

---

## [Author Response · Author response to Decision Letter 1]

18 Dec 2019

1. Please confirm that the iris thickness measurements were taken perpendicular to the posterior iris plane and include this information in the Methods of the manuscript. 

The iris thickness measurements were taken perpendicular to the posterior iris plane, we have included this information in the Methods of the manuscript. (Lines 127-128)

2. Please indicate if trabecular iris space area (TISA) measurements (mm2) were included during the study - these are usually reported for UBM studies (either at 500um or 700um from the scleral spur). If the TISA measurements were taken, please include in the manuscript and discuss the outcomes.

In patients with PCG, the dysgenesis of trabecular meshwork mainly affects the angle recess, which makes angle measurement involving the labeling of scleral spur difficult. So TISA measurement may not be accurate. On the other hand, TIA better illustrates the dysgenesis of trabecular meshwork and has been broadly used for patients with PCG. (Kobayashi H, Ono H, Kiryu J, Kobayashi K, Kondo T. Ultrasound biomicroscopic measurement of development of anterior chamber angle. Br J Ophthalmol. 1999;83(5):559-562) 

3. Line 54: please amend to 'physiological.

This has been corrected. (Line 50)

4. Line 95: 'For those with unilateral disease, the unaffected contralateral eyes served as the control group (Group 2).'

We have corrected this sentence. (Line 89-90)

5. Line 175: 'Interobserver agreement was calculated by comparing initial values of Observer 1 (YS)

to those of Observer 2 (CX).'

We have corrected this sentence. (Lines 163-164)

5. Line 382: 'Analysis of the intraobserver and interobserver reproducibility was high.'

We have corrected this sentence. (Line 355)

7. Line 390: 'Fourthly,...'

This has been corrected. (Line 363)

Reviewers' comments:

Reviewer #2: I thank the authors for their comprehensive response. I have a few minor comments below.

Line 127: I presume that the iris thickness was measured perpendicularly to the retroiridal iris plane?

The iris thickness measurements were taken perpendicular to the posterior iris plane, we have included this information in the Methods of the manuscript. (Lines 127-128)

My point about trabecular-iris angle was related more to why trabecular-iris space area was not used to account for iris surface anatomy. TISA is commonly used and I wonder why it has been omitted here.

See above.

Line 359: spelling error

This has been corrected. (Line 363)

With regard to intraocular pressure, how do the authors explain the big difference in pressure between Groups 1 and 2 even though there was no relationship found cross sectional area of Schlemm's canal (Figure 3)? Could it be related to the degree to which the canals were closed/narrow?

We agree with the reviewer that the IOP may affect the cross-sectional area of Schlemm’s canal (SC) as previous study revealed in POAG (Mu L, Yin Z, Yan X, Hong Z. The Relationship between the 24-hour Fluctuations in Schlemm’s Canal and Intraocular Pressure: An Observational Study using High-Frequency Ultrasound Biomicroscopy. Curr Eye Res. 2017;42(10):1389-1395.) But in this study, IOP was not related with the size of the largest-measured SC area in pediatric patients with PCG. One explanation is that patients with PCG have maldeveloped SC, which could be segmental. The degree of the maldeveloped SC may lead to elevated IOP, which we don’t have enough data in this study to prove this. We only performed UBM at 4 o’clock, not 360 degree. The future development of circumference SC image would help solve this question. We have discussed it in the Discussion section. (Lines 296 to 303)

---

## [Editor Report · Decision Letter 2]

26 Dec 2019

Disease-related and age-related changes of anterior chamber angle structures in patients with primary congenital glaucoma: An in vivo high-frequency ultrasound biomicroscopy-based study

PONE-D-19-17802R2

Dear Dr. Wang,

We are pleased to inform you that your manuscript has been judged scientifically suitable for publication and will be formally accepted for publication once it complies with all outstanding technical requirements.

With kind regards,

Michele Madigan

Academic Editor

PLOS ONE
---

## [Editor Report · Acceptance letter]

9 Jan 2020

PONE-D-19-17802R2 

Disease-related and age-related changes of anterior chamber angle structures in patients with primary congenital glaucoma: An in vivo high-frequency ultrasound biomicroscopy-based study 

Dear Dr. Wang:

I am pleased to inform you that your manuscript has been deemed suitable for publication in PLOS ONE. Congratulations! Your manuscript is now with our production department. 

With kind regards,

on behalf of

Dr. Michele Madigan 

Academic Editor

PLOS ONE